# Developing Novel G-Quadruplex Ligands: From Interaction with Nucleic Acids to Interfering with Nucleic Acid–Protein Interaction

**DOI:** 10.3390/molecules24030396

**Published:** 2019-01-22

**Authors:** Zhi-Yin Sun, Xiao-Na Wang, Sui-Qi Cheng, Xiao-Xuan Su, Tian-Miao Ou

**Affiliations:** School of Pharmaceutical Sciences, Sun Yat-sen University, Guangzhou 510006, China; sunzhy0118@163.com (Z.-Y.S.); sheena_wong@163.com (X.-N.W.); 18720991247@163.com (S.-Q.C.); suxx@mail2.sysu.edu.cn (X.-X.S.)

**Keywords:** G-quadruplex, G-quadruplex ligand, G-quadruplex-related proteins, helicase, anti-tumor

## Abstract

G-quadruplex is a special secondary structure of nucleic acids in guanine-rich sequences of genome. G-quadruplexes have been proved to be involved in the regulation of replication, DNA damage repair, and transcription and translation of oncogenes or other cancer-related genes. Therefore, targeting G-quadruplexes has become a novel promising anti-tumor strategy. Different kinds of small molecules targeting the G-quadruplexes have been designed, synthesized, and identified as potential anti-tumor agents, including molecules directly bind to the G-quadruplex and molecules interfering with the binding between the G-quadruplex structures and related binding proteins. This review will explore the feasibility of G-quadruplex ligands acting as anti-tumor drugs, from basis to application. Meanwhile, since helicase is the most well-defined G-quadruplex-related protein, the most extensive research on the relationship between helicase and G-quadruplexes, and its meaning in drug design, is emphasized.

## 1. Introduction

Cancer is one of the major diseases that pose a serious threat to human life and health. Due to the complicated pathogenesis of cancer, there are still many challenges in cancer therapy, despite great efforts made in the research of anticancer drugs. Finding novel anti-tumor drugs with high selectivity and few side effects is still the main problem of anti-tumor drug research. Therefore, more and more novel targets and novel strategies are being discovered and developed.

Several traditional chemotherapeutic drugs exhibit significant efforts on both normal cells and cancer cells, since they interact directly with the duplex DNA. Developing novel drugs that interact with nucleic acids using novel strategies is a significant consideration in research. According to this, finding anti-tumor agents that target the G-quadruplex structure in nucleic acids has been raised as an alternative drug development strategy, since it might increase the selectivity and specificity of drugs on certain genome regions. The G-quadruplex is a non-classical secondary structure of nucleic acids that self-folds within a sequence containing continuous guanine (G) repeats [1]. Multiple mapping and functional studies have revealed important roles of G-quadruplex structures in the regulation of gene expression and transcription, protein translation and proteolysis, DNA repair, maintenance of the stability of chromosome ends, and epigenetic regulation [2,3,4,5,6,7]. For now, many selective G-quadruplex ligands show potential for antitumor therapy applications by causing DNA damage responses and growth arrest in human cancer cells [8,9,10,11,12]. 

The search for and further optimization of compounds targeting the G-quadruplexes may lead to compounds of increasing specificity and drug potential [13,14]. However, compared to developing inhibitors for a specific enzyme or protein, selective interaction with the G-quadruplex structures in particular genome regions is difficult to achieve. An alternative strategy for discovering novel G-quadruplex-related compounds is to interfere with the binding between G-quadruplex-forming sequences and the binding proteins [15,16,17]. Considering the fact that the shifts between various secondary structures in nucleic acids actually are regulated by several proteins binding to the nucleic acids [18,19,20,21], this alternative strategy seems attractive. Therefore, we will further discuss the proteins that interact with G-quadruplexes, including both stabilizing and dissociating proteins, based on emerging findings regarding this kind of binding proteins.

In addition, the helicases are a class of molecular motor proteins that unwind DNAs or RNAs using the energy produced by the hydrolysis of nucleotide triphosphates (NTP) [22]. Helicases play essential roles in nucleic acid metabolism by facilitating cellular processes including replication, recombination, DNA repair, and transcription [23,24,25]. Furthermore, several members of the helicase family have the ability to regulate the degradation of G-quadruplexes, and subsequently regulate related biological processes to achieve anti-cancer effects [26,27,28,29,30,31,32]. Small molecules with influence on the function of G-quadruplexes via helicase have been discovered, developed, and well-evaluated [33,34]. Therefore, we hope to focus on the progress made in helicase-related leading compounds to give a comprehensive view of this field.

## 2. G-Quadruplexes 

A G-tetrad structure in guanylic acid, according to X-ray diffraction data, was first reported by Gellert et al. in 1962 [35]. Twenty years later, studies showed that the G-quadruplex structure can form in G-rich repeats at the ends of telomeres [36]. By immunostaining the telomeric G-quadruplexes using a specific antibody, G-quadruplex structures were proven in 2001 to be formed in cells [37]. From then on, several G-quadruplex-specific single-chain antibodies (scFv antibody) have been developed using different display processes, and these scFv antibodies have been used in DNA or RNA G-quadruplex mapping in cells [38,39,40]. Combined with next-generation sequencing, genome mapping and thorough functional elucidation have been reported [3]. All these results support the existence of G-quadruplex structures in the genome.

The G-quadruplex structure is a stacked secondary structure that can form in a specific repetitive sequence of G-rich DNA or RNA. The core structures in the G-quadruplex are two or three G-quartets, which form from four guanines via Hoogsteen hydrogen bonds. In addition, the G-quadruplex is stabilized by univalent metal cations (Na^+^ or K^+^) located in the central channel of the plane (Figure 1). G-quadruplex structures may form intramolecular G-quadruplexes within a single DNA strand, or intermolecular G-quadruplexes between multiple DNA strands [41]. Moreover, due to the various orientations of nucleic acids strands during folding, the G-quadruplex structures can further divide into different conformations, including parallel, antiparallel, and hybrid conformations (Figure 2). Generally, the configuration and stability of the G-quadruplex structures are related to the length and the composition of the G-quadruplex-forming sequence, the length of the annulus structure between Gs, the number of DNA strands, and the type of binding cations [42,43,44,45].

Recent studies show that G-quadruplexes are involved in multiple cellular events, including DNA replication [54,55,56], DNA damage repair [29,57,58], transcription [59,60,61,62,63], RNA processing [4,64,65,66], translation [67,68,69], and epigenetic regulation [7,70]. G-quadruplexes can block the fork process and thus inhibit gene replication during mitosis (Figure 3a), and also play a role in the inhibition of DNA damage repair (Figure 3b). G-quadruplexes located upstream or downstream of the transcription start site (TSS) can inhibit or promote transcription (Figure 3c). In addition, the formation of G-quadruplexes can recruit certain translation initiation proteins or block these proteins’ binding to the untranslated region (UTR), and thus have an influence on translation (Figure 3d).

Next, we will discuss the telomere G-quadruplex, DNA G-quadruplex, and RNA G-quadruplex, independently.

### 2.1. The Telomere G-Quadruplex

The telomeres, located at the ends of eukaryotic chromosomes, protect the chromosomes from degradation and recombination due to faulty DNA repair signals [71]. Eukaryotic chromosomes become shorter and shorter during replication with cell division, which eventually leads to cell senescence and apoptosis [72]. In most eukaryotes, the telomeres recruit telomerase to compensate for cellular damage [73]. Specifically, telomere DNA exists as a single-stranded overhang and serves as the substrate for reverse transcription catalyzed by the telomerase. Once a G-quadruplex structure forms in this single-stranded DNA, the activity of the telomerase in this process is inhibited [74]. This inhibitory activity can be further reinforced by stabilizing the G-quadruplex structure via specific small molecules [75,76]. In addition, the G-quadruplex affecting telomerase recruitment is also regulated by many other binding proteins and helicases, such as the protection of telomeres 1 (POT1) [77,78], the telomere binding protein TRF1 [79] and TRF2 [80,81,82], and the heterogeneous nuclear ribonucleoprotein A1 (hnRNP A1) [83,84].

On the other hand, G-quadruplexes formed in the telomere RNA (TERRA) can also affect chromosome elongation, which forms an antiparallel RNA G-quadruplex with rGs, adopting a *syn/anti* conformation [85]. Actually, since multiple G-repeats exist, there is more than one quadruplex in the telomeric single-stranded overhand, including the DNA quadruplexes stacking to form a higher-order quadruplexes [86], DNA:RNA hybrid quadruplexes [87,88], and RNA quadruplexes stacking to dimer quadruplexes [85]. The shift between different secondary structures might also be regulated by binding proteins, for example, hnRNPA1 can bind to and dissociate RNA telomere G-quadruplexes [89].

Formation of telomere–G-quadruplexes is closely related to tumorigenesis. Therefore, targeting telomere–G-quadruplexes becomes a promising anti-tumor strategy [14]. The first reported telomere–G-quadruplex ligand was found in 1997 [90], which can inhibit the elongation of the telomere by the telomerase. The researchers then successfully developed a large number of compounds with potential anti-tumor activities targeting telomere–G-quadruplexes [76].

### 2.2. DNA G-Quadruplexes

Despite of the existence of G-quadruplexes in telomere DNA, there are over 700,000 G-quadruplex-forming sequences in the human genome [91]. More importantly, most of these sequences are in functional regions, including the telomere end discussed above, the promoter regions of oncogenes, ribosomal DNA, the 5′ untranslated region (5′-UTR) in mRNAs, and so on.

Most of quadruplex-forming sequences exist in the gene promoter regions. Several studies have revealed the extensive presence of the G-quadruplex in the promoter region, and suggested that the G-quadruplex may regulate gene transcription [92,93,94,95,96,97].

The first reported G-quadruplex in the promoter region is formed in the nuclease hypersensitivity element III_1_ (NHE III_1_) of the proto-oncogene *c-myc* [59,94] NHE III_1_ locates upstream of the promoter 1 (P1) of the *c-myc*, and is responsible for most of the transcription regulation of the gene [98]. The G-rich sequences in this region has well been studied and shown to form a parallel G-quadruplex [99]. In addition to the *c-myc*, there are many G-quadruplexes that have been proven to form in the promoter regions, such as proto-oncogenes *VEGF* [97,100], *bcl-2* [95,101], *c-kit* [102], *HIF-1* [103], *RET* [104], and *PDGF-A* [105]; DNA repair gene *RAD17* [106]; the human platelet-derived growth factor receptor beta *PDGFR-β* [96,107]; the homeobox gene *HOXC10* [108]; the androgen receptor gene *AR* [109,110]; or human myosin gene (*MYH7*) [111]. The formation of G-quadruplexes in these promoter regions hinders the interactions between DNA and its transcription factors, which in turn regulate the transcription. Multiple studies have shown that G-quadruplex ligands can reduce the expression of these genes, indicating that the presence of the G-quadruplex structure might act as a switch in gene transcription [100,112,113,114,115,116,117].

The formation of G-quadruplexes can not only inhibit the transcription process, but also promote the transcription in some genes [118,119,120]. This needs to be discussed in different situations. G-quadruplexes located upstream of the transcription start site (TSS) can inhibit transcription when the formation interferes with the binding of the RNA polymerase II or transcription factors [121], while it can promote transcription initiation when it recruits specific transcription factors to the single-stranded region [119,120]. On the other hand, G-quadruplexes formed downstream of the TSS of the template chain can hinder the recognition of the RNA polymerase II, and thus lower the transcription level. However, G-quadruplexes locating in the coding chain under this situation can interrupt the transcriptional product at the position, and thus inhibit the transcriptional process [122]. Although the effects of G-quadruplexes on transcription seem complicated, these different functions are all achieved by different kinds of nucleic acid binding proteins.

Therefore, the nucleic acid binding proteins and their role in G-quadruplex-related regulation are more and more attractive. Analyses of the human genome find that the binding sites of helicase, XPB and XPD, overlap the site of G-quadruplex formation in the promoter regions [32]. These two helicases can be recruited into the G-quadruplex-forming region and unwind the G-quadruplex structure, so that the transcription can proceed smoothly. Interestingly, G-quadruplexes exist not only in the promoter region but also at the end of the gene, which suggests that G-quadruplexes affect not only the initiation, but also the termination of gene transcription [123].

Analyses of genome initiation sites imply that the G-quadruplexes also play a role in the DNA replication and modification [124,125]. For example, 35% of replication initiation depends on the CpG island, and the G rich sequence around CpG island is actually very frequent, with a high distribution rate of up to 80% [126,127]. Moreover, the presence of G-quadruplex structures is associated with CpG island hypomethylation in the human genome, via inhibition of DNA methyltransferase 1 (DNMT1) enzymatic activity [7].

Since G-quadruplexes have been shown to cause genomic instability, the effect of the G-quadruplex on DNA damage repair is to elicit a DNA damage response by causing the formation of DNA double strand breaks (DSB) [128]. Specialized helicases that unwind G-quadruplexes have been shown to prevent genetic instability [129]. For example, when FANCJ is missing, a single unresolved G-quadruplex structure can persist through multiple mitotic divisions, which might increase the risk of DNA double-strand breaks [130].

### 2.3. RNA G-Quadruplexes

RNA G-quadruplexes have also been recently shown to have various regulatory activities. Recent methodological developments, including predictive algorithms and structure-based sequencing, have made it possible to detect and map RNA G-quadruplex structures in transcriptomes on large scales, with high sensitivity and resolution [131]. RNA G-quadruplexes are thought to play a key role in many biological processes, such as transcription and post-transcriptional events [132].

Multiple genomic studies have indicated that the majority of RNA G-quadruplexes form in the untranslated region (UTR) of mRNA, which recruits translational proteins and regulates translation [4]. In fact, in vitro experiments have revealed that G-quadruplexes participate in the regulation of gene translation [133]. For example, the 5’-UTR of the oncogene *NRAS* contains typical G-quadruplex-forming sequence, and the formation of the structure can inhibit gene expression [67,134]. The formation of G-quadruplexes in the 5’-UTR of proto-oncogene *VEGF-A* can regulate cap-independent translation initiation [135,136]. A translational protein, eIF4A, can recognize the repeat sequence of CGG in the UTR, and accelerate the progress of T cell acute lymphoblastic leukemia by unwinding the G-quadruplexes in this repeat [30].

In addition to the UTR of mRNA, quadruplexes in alternative splicing (AS) sites might act as cis-elements to regulate the post-transcription process [137]. For instance, G-quadruplex forming in the sixth intron of h*TERT* gene acts as an intron splicing silencing element and reduces the splicing efficiency [138]. A G-quadruplex ligand, CX-5461, seems to be able to regulate AS in h*TERT*, showing therapeutic potential for glioblastomas [65]. In contrast, G-quadruplex formation in the third intron of the *TP53* gene promotes the splicing of intron 2 [139,140]. In addition to G-quadruplexes in introns being able to regulate AS, G-quadruplexes located in exons can also regulate AS. For example, two G-quadruplexes in the 15th exon of fragile mental disorder gene *FMR1* have been shown to enhance efficiency of splicing [141], and the production of splicing products of the *FXYD1* and *TR12* genes was also regulated by G-quadruplexes [142,143]. The reason why G-quadruplex structure can regulate AS may be that purine splicing regulation sequence influences splicing enhancement by interacting with specific splicing proteins to enhance efficiency [144].

## 3. G-Quadruplex Interacting Compounds

G-quadruplexes show a wide range of biological functions, including telomere maintenance, transcription, translation, replication, DNA damage response, genome rearrangement, and epigenetic regulation [2]. Therefore, designing small molecules that interact with G-quadruplexes might help to find novel compounds with anti-tumor activities. Over the past 20 years, various small molecules that interact with either DNA G-quadruplexes or RNA G-quadruplexes have been reported, some of which show potential anti-tumor activities.

According to the different biological functions of G-quadruplexes in different regions, molecules interacting with quadruplexes can influence cells in different ways: (1) suppression of oncogenes’ expression by stabilizing DNA G-quadruplexes in the promoters [8,145]; (2) small molecules inhibiting telomerase activity and eliminating the unlimited proliferation of tumor cells by stabilizing G-quadruplexes at the end of chromosomes [146]; and (3) blocking replication forks and inducing ssDNA gaps or breaks in tumor cells [128,147].

G-quadruplex ligands are small molecules that can bind to G-quadruplexes with high affinity. In general, the binding constant (*K*_D_) between ligands and G-quadruplexes is lower than 10^−6^ mol·L^−1^. The patterns by which small molecule can bind to G-quadruplexes are stacking with the outer G-quartets, groove binding, loop binding, and combined binding [148,149]. According to these binding modes, there are several common structural characters of G-quadruplex ligands, including a polycyclic heteroaromatic core that can be combined with G-quadruplexes, and some charged hydrophilic groups to facilitate binding to G-quadruplex grooves and loops. At the same time, these ligands should be stable under physiological condition. Furthermore, the druggability of compounds also needs to be considered [150].

Basing on the above characteristics, different types of small molecules have been reported. Since we hope to focus our discussion on the anti-tumor potential of G-quadruplex ligands, we will next emphasize several typical compounds with significant biological activities, especially anti-tumor activities evaluated in vivo (Table 1 and Figure 4).

### 3.1. The 3,6,9-Trisubstituted Acridine Derivative BRACO19 and Other Acridine Derivatives

BRACO-19 was optimized from disubstituted acridine derivative, and was first reported as a telomerase inhibitor with an IC_50_ value of 115 nM [151]. BRACO-19 can bind to the telomeric G-quadruplex via three binding modes, top stacking, bottom intercalation, and groove binding [171]. The mechanism and anti-tumor activity of BRACO-19 are well studied. In brief, BRACO-19 can uncap 3’ telomere ends [152], inhibit the helicase activity of BLM and WRN proteins on G4 and B-form DNA substrates [172], and trigger extensive DNA damage response at the telomeres [153]. **B** BRACO-19 shows good anti-tumor activities alone or in combination in several human cancers, including anti-tumor activity on human epidermoid carcinoma A431 cells [151], flavopiridol-resistant colorectal cancer HCT-116 cells [154], human uterus carcinoma UXF1138L cells [155], human prostate cancer DU145 cells [156]. Recently, BRACO-19 has been further developed as anti-HIV agents [173,174]. However, the very poor permeability of BRACO19 limits its further development, and further application requires a suitable formulation to ensure adequate delivery across cellular barriers [175].

On the basis of BRACO-19, further optimizations of 3,6,9-trisubstituted acridine compounds were done with systematic variations at the 3-, 6-, and 9-positions [176,177]. Long-term exposure of human breast cancer MCF7 cells to a subset of the most active compounds showed that one compound produced a marked decrease in population growth, accompanied by senescence [176]. Trisubstituted acridine–peptide conjugates and triazole–acridine conjugates were also designed to increasing the ability to recognize and discriminate between various DNA quadruplexes. The conjugates displayed quadruplex affinities in the 1–5 nM range, and at least 10-fold discrimination between the quadruplexes [178,179].

In addition, similar acridine derivatives, including bis(quinacridine) macrocycle [174], dibenzophenan-throlines [180,181], mono- and bis-pyrimidinoacridines [182], 4,5-bis(dialkylaminoalkyl)-substituted acridines [183], and 5,6-dihydrobenzo[*c*]acridine [184,185], also showed stabilizing effects on G-quadruplexes and high telomerase inhibitory activity, due to the structural similarity with the G-quartet.

### 3.2. The Cationic Porphyrins TMPyP4 and Metallo-Organic Compounds Derived from Porphyrin

Considering the similarity between the G-quartet and the porphyrin scaffold, cationic porphyrins were designed and identified as strong G-quadruplex ligands. The most typical example is TMPyP4 [157]. This compound shows high affinity with G-quadruplexes, good inhibitory activity on telomerase, and inhibitory activity on the expression of oncogenes (such as *c-myc*, *k-Ras, bcl-2,* or *c-met*) [59,60,186,187,188,189,190]. TMPyP4 shows anti-tumor activity in several tumor cells, including (1) retinoblastoma cell lines, by inducing *p53* expression and activating p38 in the MAPK–JNK–ERK signaling pathway [161]; (2) leukemia cell lines, by reducing *c-myc* expression and promoting the p21^CIP1^ and p57^KIP2^ proteins to activate p38 [160]; (3) prostate carcinomas, by downregulating *c-myc* expression and inhibiting telomerase activity [159]; (4) melanoma cells, by decreasing *RAS* expression in the ERK pathway [10]. Therefore, TMPyP4 has broad prospects in the field of tumor treatment.

In addition, the binding modes of TMPyP4 with different G-quadruplexes are also well studied [191,192,193]. Therefore, in addition to use as an anti-tumor compound, TMPyP4 is also used as a tool and a probe for G-quadruplex-related studies.

At the same time, a core-modified expanded porphyrin analogue, Se2SAP, was designed and synthesized. Se2SAP converts the parallel *c-myc* G-quadruplex into a mixed parallel/antiparallel G-quadruplex with one external lateral loop and two internal propeller loops, resulting in strong and selective binding to the G-quadruplex [194]. Se2SAP shows stronger interaction ability on G-quadruplex than TMPyP4, and suppresses *VEGF* transcription in different cancer cell lines, including HEC1A and MDA-MB-231 [100].

To mimic the stabilization effect of K^+^ or Na^+^ in the central anion channel of the G-quadruplex, introducing positive charges to the aromatic core seems to be an attractive strategy. Thus, introducing *N*-methylated modification to mimic the interactions between cations and the anion central channel is a common strategy for G-quadruplex ligand optimization. Both TMPyP4 and Se2SAP possess *N*-methylated groups. On the other hand, designing metallo-organic compounds can also directly improve the interactions between chelators and G-quadruplexes. The cationic or highly polarizing properties of these metallo-organic compounds are also significantly conducive to promoting their binding to the negatively charged G-quadruplexes. Inserting metal into the center cavity of the TMPyP4 can result in Ni(II)-, Mn(III)-, or Cu(II)-complexes, which show good stabilization activity on G-quadruplexes [195]. The inhibition activity on telomerase of Mn–TMPyP4 was less than that of TMPyP4, but it showed an around 10-fold preference for G-quadruplex over duplex DNA [196].

### 3.3. Natural Macrocyclic G-Quadruplex Ligands: Telomestatin

Telomestatin is a typical example of a natural macrocyclic compound. It was isolated from *Streptomyces annulatus* in 2001 and has been widely studied [197,198,199]. Its strong telomerase inhibition makes it a research hotspot. It inhibits tumor cell proliferation by changing telomere conformation and length, and by dissociating the telomere-binding proteins. As a result, telomestatin showed good cytotoxicity and induced apoptosis in different types of tumor cells, while it had no effect on normal cells. At the same time, reduced telomerase activity, shortening telomeric length, activation of the DNA damage response related to ATM kinase, and increased expression of p21 and p27 can be observed in human leukemia cell line K562 under treatment with telomestatin [163]. Recent in vivo data revealed that telomestatin potently eradicates glioma stem cells (GSC) through telomere disruption and *c-Myb* inhibition, suggesting a novel GSC-directed therapeutic strategy for glioblastoma multiforme (GBM) [162]. On the basis of telomestatin, some analogues of telomestatin, such as HXDV and 6OTD, were synthesized and showed strong inhibition on tumor cells with no effect on either duplex or triplex DNA [200,201,202].

### 3.4. Pyridine Derivative Pyridostatin and Its Analogues

Pyridostatin (PDS) was rationally designed according to certain structural features shared by known quadruplex-binding small molecules, with particular emphasis on an electron rich aromatic surface, the potential for a flat conformation, and an ability to participate in hydrogen bonding [167]. PDS increases telomere fragility in BRCA2-deficient cells via stabilization of the G-quadruplex in the telomeric region, and thus reduces proliferation of homologous recombination (HR)-defective cells by inducing double-strand breaks accumulation and checkpoint activation, and deregulating G2/M progression [168].

PDS is also a widely-used probe for G-quadruplexes. For example, combining RNA G-quadruplex sequencing (rG4-seq) and PDS on polyadenylated-enriched HeLa RNA helped generate a global in vitro map of rG4 formation and uncover rG4-dependent differences in RNA folding [4]. Moreover, a synthetic small molecule derived from an N,N’-bis(2-quinolinyl)pyridine-2,6-dicarboxamide containing an affinity tag is used to mediate the selective isolation of G-quadruplex nucleic acids [203]. A cross-linking agent that combines the nitrogen mustard chlorambucil with PDS can alkylate G-quadruplex structures and selectively impair growth in cells genetically deficient in nucleotide excision repair (NER) [204].

### 3.5. Fluoroquinolone Antibiotics CX-3543 and CX-5461

CX-3543 (also known as quarfloxin) is a fluoroquinolone derivative, and the first G-quadruplex interactive agent to enter human clinical trials. It binds to G-quadruplex DNA and has been shown to selectively disrupt interaction of rDNA G-quadruplexes with the nucleolin protein, thereby inhibiting Pol I transcription and inducing apoptotic death in cancer cells [169]. Another compound possessing a similar mechanism is CX-5461, a potent small-molecule inhibitor of rRNA synthesis in cancer cells that selectively inhibits Pol I-driven transcription DNA replication and protein translation. CX-5461 is orally bioavailable, and demonstrates in vivo anti-tumor activity against human solid tumors in murine xenograft models [170]. Therefore, this drug is now in advanced phase I clinical trial for patients with BRCA1/2 deficient tumors (Canadian trial, NCT02719977, opened May 2016). Further mechanism study revealed that CX-5461 blocks replication forks and induces ssDNA gaps or breaks, which need the BRCA and NHEJ pathways for repair [147].

## 4. G-quadruplexes and Their Binding Proteins

More and more studies have suggested that G-quadruplexes could not regulate biological processes by themselves. Various proteins take part in this regulation. Proteins interacting with G-quadruplexes can be divided into three types according to their effects on G-quadruplexes: promoting G-quadruplex formation or stabilizing G-quadruplexes, unwinding G-quadruplexes, and degrading G-quadruplexes. A selection of functional proteins is summarized in Table 2.

### 4.1. Proteins Promoting G-Quadruplex Formation

Among the proteins promoting G-quadruplex formation, nucleolin (NCL) is the most commonly reported. It is widely believed that NCL plays a partner role by helping the correct folding of complex nucleic acid structures. NCL is a nucleolar phosphorylated protein highly expressed in proliferating cells, which plays an important role in ribosomal biogenesis [205], chromatin remodeling [206], and transcription [207]. NCL can bind to and promote *c-myc* G-quadruplex structures in vitro with high affinity and selectivity [208]. In addition, NCL is able to bind specifically to the promoter region of the *VEGF* gene in negatively supercoiled DNAs [209]. Additionally, NCL also plays a role in promoting G-quadruplex formation in viral genomes. It is able to specifically recognize G-quadruplex structures present in the HIV-1 LTR promoter or Epstein–Barr virus, and increase promoter silencing activity [210,211]. According to these findings, a quadruplex-forming oligonucleotide aptamer, AS1411, is currently in clinical trials as a treatment for various cancers by affecting the activities of certain NCL-containing complexes [212,213,214].

Human topoisomerase plays a crucial role in DNA replication, transcription, and chromosome condensation. Several topoisomerases (Topo), such as Topo1, Topo1b, and Topo2, bind specifically to pre-formed parallel and anti-parallel G-quadruplexes, and are able to promote the formation of these structures [215,216,217,218].

Unlike NCL and Topo, the effect of the cellular nucleic-acid-binding protein (CNBP) on G-quadruplexes is not very clear. It might facilitate the formation of G-quadruplexes in the NHE III_1_ region of gene *c-myc* and thus activate transcription [219]. Studies in *Bufo arenarum* have indicated that the promoting function of the CNBP might due to its binding to RNA and single-stranded DNA. Specifically, CNBP functions as a nucleic acid chaperone through binding, remodeling, and stabilizing nucleic acid secondary structures [220,221].

### 4.2. Proteins Degrading G-Quadruplexes

Proteins that can degrade G-quadruplex DNA or RNA are not well studied, and most of them are nucleases. One such example is a human nuclease, GQN1 (G quartet nuclease 1), which cuts within the single-stranded region formed by stacked G-quartets. GQN1 degrades G-quadruplex DNA but does not degrade duplex or single-stranded DNA or G4 RNA [222]. Another case is the *Saccharomyces cerevisiae* Mre11 protein (ScMre11p), which possesses high binding affinity for G-quadruplex DNA over single- or double-stranded DNA. Binding of ScMre11p to G-quadruplex DNA or G-rich single-stranded DNA is accompanied by endonucleolytic cleavage at flanking sites of G residues and G-quartets [223,224].

### 4.3. Proteins Unwinding G-Quadruplexes: Helicase

Among these proteins, helicases are motor proteins able to unwind nucleic acids. In 1976, the first helicase, Tral (helicase 1), was found in *Escherichia coli* cells [225]. Since then, 95 helicases in human cells, including 31 DNA helicases and 64 RNA helicases, have been found [226]. They are widely involved in almost all aspects of cell metabolism: replication, repair, recombination, transcription, chromosome isolation, and telomere maintenance [22,23,227,228,229]. Although the main function of helicases is to catalyze the formation of single-stranded nucleic acids, there is growing evidence to show that some of them are involved in the active decomposition of other non-standard DNA structures, such as G-quadruplexes.

DNA helicases are divided into 6 superfamilies (SF) according to their amino acid sequences (Table 3) [24]. Depending on the direction of movement on the DNA, the helicase can also be divided into two types: type A (3’ to 5’) or type B (5’ to 3’). SF1, SF2, and SF6 superfamilies contain both type A and B helicases. So far, all of the discovered SF3 proteins are type A, and all of the members of SF4 and SF5 superfamilies belong to type B. Depending on whether the helicase moves on single-stranded or double-stranded DNA, it can also be divided into ‘α type’ and ‘β type’. So far, all SF1 enzymes seem to be α type, while SF2 superfamily include both αand β type.

Different families of helicases have different activities, but all of them share some common characteristics. For example, all G-quadruplex-helicases require a single-stranded tail on either the 3’ or 5’ end, which ensures they can be loaded onto the DNA substrate [25]. In addition, all G-quadruplex helicases use ATP hydrolytic energy to unwind G-quadruplex structures except for WRN and BLM, which are surrounded by single-stranded DNA [255,256].

The Pif1 subfamily is a class of ATP-dependent 5′ to 3′ helicases, and is widely found in prokaryotic and eukaryotic cells and viruses. All Pif1 helicases contain a conserved Pif1 domain of 300 to 500 amino acids [230]. Pif1 helicases are involved in telomere elongation, synthesis of rDNA and Okazaki fragments, and maintaining chromosome stability [231]. Specifically, human Pif1 (hPif1) unwinds DNA double strands, DNA:RNA hybrid double strands, and secondary structures to promote gene transcription in the presence of Mg^2+^ [232,257]. For example, hPif1 unwinds telomeric DNA:RNA hybrid double strands in the telomere, and inhibits telomere function in tumor cells via binding to the G-quadruplex structure in this region [232].

The RecQ subfamily is a class of helicases belonging to SF2, which is highly conserved in the evolutionary process and widely expressed in multicellular organs [238,239]. Most helicases in the RecQ subfamily contain a helicase core (RQC), a RecQ C-terminal, a helicase, and a RNaseD C-terminal (HRDC). One of helicases in this subfamily, the Bloom helicase (BLM), is the first helicase to be verified as G-quadruplex unwinding helicase [247], and can unfold both intermolecular and intramolecular G-quadruplexes [258,259]. Study on the network of mRNAs and miRNAs in BLM-deficient cells has indicated that G-quadruplex motifs are enriched at transcription start sites, and especially within first introns of differentially expressed mRNAs, in Bloom syndrome compared with normal cells, which may drive the pathogenesis of Bloom syndrome [260]. With the development of research on BLM and G-quadruplexes, more and more functions and mechanisms are revealed, including in DNA double-strand breaks repair [240], excessive sister chromatid exchange [241], and chromosomal rearrangements [242]. Another important member of this subfamily is the Werner-syndrome-associated helicase (WRN), which shows similar functions to BLM [243,244]. For example, both BLM and WRN facilitate telomere replication during leading strand synthesis of telomeres [245]. The unwinding of a G-quadruplex by BLM and WRN can be suppressed by HERC2, a HECT E3 ligase [18].

FANCJ helicase is a kind of ATP-dependent 5 ‘to 3’ DNA helicase, which is widely involved in DNA damage repair, G-quadruplex disassembly, homologous chromosome recombination, and maintaining genomic stability. In the process of replication fork formation, FANCJ promotes the partial unwinding of double-stranded DNA into a single strand, which facilitates the formation of G-quadruplexes and hinders the synthesis of DNA by DNA polymerase [235]. FANCJ can further unwind and remove the G-quadruplex structure, allowing DNA replication to proceed smoothly [26,236]. FANCJ deficiency will stop replication at the G-quadruplex forming site, and will eventually cause DNA damage [27,236].

RHAU (an RNA helicase associated with the AU-rich sequence of mRNAs) is the product of gene *DHX36*, and is also named G4 Resolvase 1 (G4R1). It binds to and resolves tetramolecular RNA as well as DNA quadruplex structures [31,251,261]. RHAU is a multi-functional helicase that has been implicated in G-quadruplex-mediated transcriptional and post transcriptional regulation, and is essential for heart development, hematopoiesis, and embryogenesis in mice [31,252,253,262]. A co-crystal structure of bovine RHAU bound to DNA with a G-quadruplex and a 3’ single-stranded DNA segment shows that the N-terminal RHAU-specific motif folds into a DNA-binding-induced alpha-helix that selectively binds parallel G-quadruplexes. G-quadruplex binding alone induces rearrangements of the helicase core to drive G-quadruplex unfolding one residue at a time [19].

Many G-quadruplex structures have high thermal stability compared to double-stranded or single-stranded DNA, thus helicases facilitate the maintenance of a balance between different secondary structures.

Due to the widespread existence of G-quadruplex-forming sequences in the genome and their structural polymorphism, it is not very easy to discover G-quadruplex ligands with absolute specificity. Alternatively, interfering with the binding or interaction between G-quadruplexes and helicases shows their biological relevance.

#### 4.3.1. Effects of G-Quadruplex Ligands on Quadruplex-Related Proteins

As mentioned above, telomestatin and TMPyP4 are two typical stabilizers of G-quadruplex structures, and have been well and widely studied. The effects of these compounds on quadruplex-related proteins have also been studied, since they are usually used as probes. Telomestatin can reduce the expression of telomere-binding protein in HeLa cells, leading to dissociation of the TRF2 from the telomere and eventually to disorder in telomere functions [263]. Therefore, exposure of human tumor cells to telomestatin induces the dissociation of shelterin proteins, such as POT1 and TRF2, or telomere-associated proteins (e.g., topoisomerase III (TOP3)) from their telomeric sites [199,263,264]. Moreover, it has been proposed that telomestatin competes with proteins for binding to G-quadruplex DNA or stabilizing a G-quadruplex structure that is not favorably bound by the telomere-interacting protein, leading to telomere uncapping [199,264]. At the same time, FANCJ is involved in this process, since G-quadruplex is a physiological substrate of FANCJ [265]. Telomestatin-treated FANCJ-depleted cells showed impaired proliferation, apoptosis, and increased DNA damage levels [235,266].

TMPyP4 shows inhibitory activity on telomerase, and can cause cell arrest in S and G2/M phases [267,268]. It shows inhibition activity of RecQ helicase unwinding activity on G-quadruplex DNA, such as *E. Coli* RecQ helicase [269] and *S.cerevisiae* Sgs1p helicase [193]. TMPyP4 also exacerbates telomere fragility, in which TRF1 acts suppressor by recruiting RTEL1 and BLM [71]. In addition, a similar structure to TMPyP4, *N*-methyl mesoporphyrin IX (NMM) (Figure 5), serves as a specific G-quadruplex-related helicase inhibitor. When NMM exists, the helicase (such as BLM and Sgs1p) is trapped by the NMM–G-quadruplex complex without unwinding [269].

On the other hand, a polyxylene derivative, PIPER (Figure 5), is able to specifically inhibit *S.cerevisiae* Sgs1p helicase’s unwinding of G-quadruplex structure, with no effect for double-stranded DNA [246]. On the basis of PIPER, a series of analogues were synthesized, among which Tel11 has shown a strong selectivity to inhibit the unwinding activity of T-ag. Tel11 is an effective G-quadruplex helicase inhibitor of SV40 T-antigen, which binds to the substrate DNA by high stoichiometry and slowly separates from the complex [248].

#### 4.3.2. Ligands Designed to Block Protein–G-Quadruplex Interactions

The metastasis suppressor gene *NME/nm23/NDPK* was discovered in 1988 [270]. Several lines of evidence implicate the role of NM23 proteins in transcriptional regulation of gene expression [271]. Importantly, transcriptional activation of the *c-myc* oncogene by NM23-H2, one of the nucleoside diphosphate kinases in this family, was shown in human as well as murine cells, including in human cervical, lung carcinoma, and Burkitt lymphoma, by multiple research groups [272,273,274]. NM23-H2 is a G-quadruplex binding protein, and interaction between them can regulate gene transcription, including c-myc, PDGF-A, and the Alzheimer associated amyloid-β peptide (APP) [275,276,277,278,279]. Therefore, finding and designing small molecular ligands that can effectively block the interaction between NM23-H2 protein and DNA may become a novel anti-tumor strategy. According to this, an isaindigotone derivative, SYSU-ID-01, was verified as a blocker for NM23-H2–G-quadruplex interaction from screening [15]. SYSU-ID-01 binds to the NM23-H2 protein with little binding affinity to G-quadruplex DNA. Subsequently, the research group modified this structure to reduce the ability to stabilize G-quadruplexes, and obtained compound 37 (Figure 5), with a selective binding ability to the NM23-H2 protein and subsequent anti-tumor activity. Compound 37 is well-fitted into the narrow, slightly curved pocket that the dinucleotide possesses, and undergoes hydrogen bonding with residues in the channel of the protein active site (Gly113 and Asp121), hydrophobic interactions with His118 and Lys66, and π−π stacking with Phe60 and Tyr67 [17]. On the other hand, other isaindigotone derivatives developed by the same group were found to bind to both NM23-H2 and the G-quadruplex, and showed remarkable abilities in disrupting G-quadruplex–NM23-H2 interactions. They exhibited significant effects on *c-myc*-related processes in SiHa cells, including inhibiting transcription and translation, inhibiting cellular proliferation, inducing apoptosis, and regulating cell cycle [16].

#### 4.3.3. Direct Inhibitors for G-Quadruplex-Related Proteins

A small molecule from the National Cancer Institute Diversity Set, designated NSC 19630 (Figure 5), was identified, which inhibited WRN helicase activity but did not affect other DNA helicases, including BLM, FANCJ, RECQ1, RecQ, UvrD, and DnaB. Subsequently, exposure of human cells to NSC 19630 dramatically impaired growth and proliferation, induced apoptosis in a WRN-dependent manner, resulted in elevated γ-H2AX and proliferating cell nuclear antigen (PCNA) foci, and sensitized the cells to the G-quadruplex-binding compound telomestatin or a poly (ADP ribose) polymerase (PARP) inhibitor [33].

A high-throughput screening for BLM inhibitors was performed using 350,000 compounds from the Molecular Libraries Small Molecule Repository library in 2013. The compound MLS000559245 was selected and further modified to ML216 (Figure 5) as a lead compound. ML216 shows cell-based activity and can induce sister chromatid exchanges, enhance the toxicity of aphidicolin, and exert antiproliferative activity in cells expressing BLM [280].

Porphyrin scaffolds seem to be a promising core for helicase inhibitors. Although there is no evidence on the relevance to G-quadruplexes, a bismuth porphyrin complex (Figure 5) exhibits activities against both SARS-CoV (severe acute respiratory syndrome coronavirus) helicase, and duplex-unwinding activities through Bi–S bonds, indicating the potential application of bismuth drugs in the antiviral field [281].

## 5. Conclusions

The G-quadruplex structure is an important secondary structures of nucleic acids. The widespread existence in vital regulatory genome regions and a series of reported biological functions make this structure a promising drug target in anti-tumor drug discovery. In this review, we discuss the structures, existence, and functions of G-quadruplexes. Basing on this, we summarize some typical G-quadruplex ligands with promising anti-tumor activities. Since G-quadruplexes exert their regulatory functions mainly through the binding proteins of multiple nucleic acids, especially the helicases, we further introduce some G-quadruplex-related proteins, especially the helicase. The fact that designing molecules to block the interactions between nucleic acids and proteins is feasible makes this novel anti-tumor strategy more and more attractive.

## Figures and Tables

**Figure 1 molecules-24-00396-f001:**
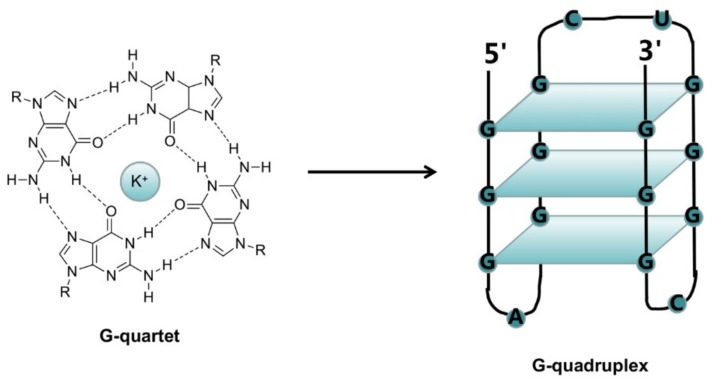
The structure of the G-quadruplex. Four guanines construct a G-quartet via Hoogsteen hydrogen bonds. Two or three G-quartets stack to form a G-quadruplex structure. Univalent metal cations (Na^+^ or K^+^) locate in the central channel of the G-quartet to stabilize the structure.

**Figure 2 molecules-24-00396-f002:**
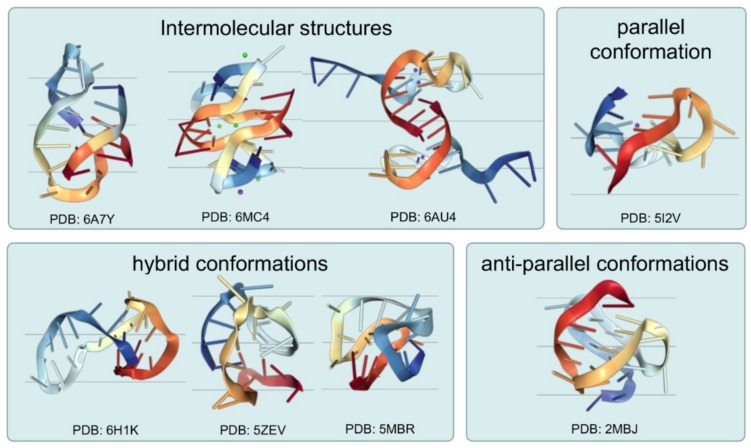
Structural diversity of G-quadruplexes. G-quadruplexes structures may form intramolecular G-quadruplex or intermolecular G-quadruplex structures (PDB: 6A7Y [46], 6MC4 [47], and 6AU4 [48]). Moreover, the G-quadruplex structures divide into different conformations, including parallel (PDB: 5I2V [49]), antiparallel (PDB: 2MBJ [50]), hybrid (PDB: 6H1K [51], 5ZEV [52], and 5MBR [53]) conformations.

**Figure 3 molecules-24-00396-f003:**
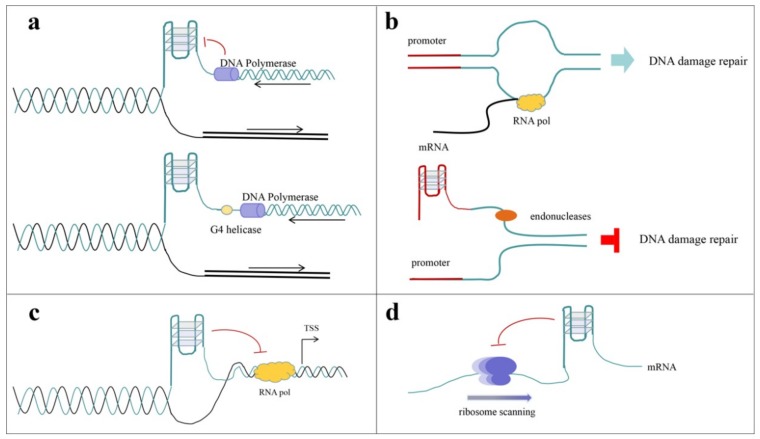
Schematic diagram of the possible role of G-quadruplexes in several cellular events. (**a**) G-quadruplexes block the replication process, and G4 helicase could withstand this inhibition. (**b**) G-quadruplexes forming in the promoter regions could interfere with the DNA damage response. (**c**) G-quadruplexes upstream of the TSS could inhibit the transcription process. (**d**) G-quadruplexes could also interfere with the ribosome scanning process and thus inhibit protein translation.

**Figure 4 molecules-24-00396-f004:**
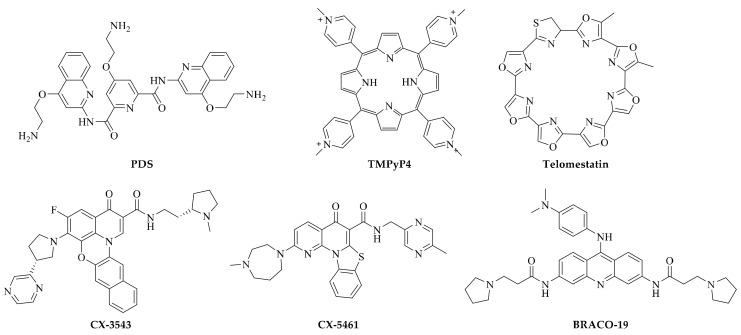
Structures of typical G-quadruplex ligands in this review, including BRACO-19, TMPyP4, Telomestatin, CX3543, CX5461, and pyridostatin (PDS).

**Figure 5 molecules-24-00396-f005:**
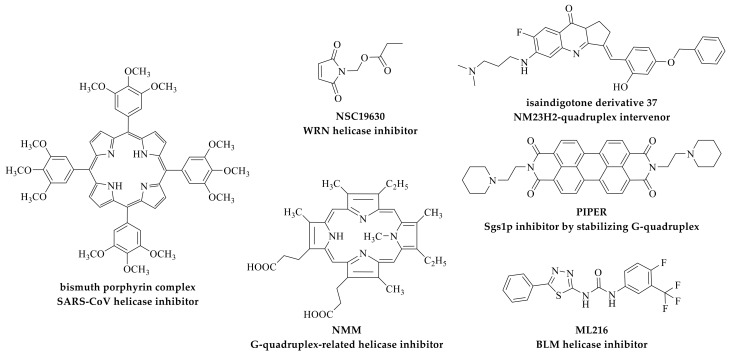
Structures of compounds inhibiting G-quadruplex-related proteins, including *N*-methyl mesoporphyrin IX (NMM), PIPER, isaindigotone derivative 37, NSC 19630, ML216, and bismuth porphyrin complex.

**Table 1 molecules-24-00396-t001:** Typical G-quadruplex ligands and their biological activities.

Compound	Biological Activities	*In Vivo* Activities
BRACO-19	Telomerase inhibition [151], uncapping of 3’ telomere ends [152], triggering extensive DNA damage response at telomere [153].	Anti-tumor activity on human epidermoid carcinoma A431 cells [151], flavopiridol-resistant colorectal cancer HCT-116 cells [154], human uterus carcinoma UXF1138L cells [155], and human prostate cancer DU145 cells [156].
TMPyP4	Telomerase inhibition and shortening the telomere length [157], promoting the formation of both G-quadruplex and i-motif [158], inhibiting oncogene transcription [159].	Anti-tumor activity on PC-3 human prostate carcinomas [159], K562 leukemic cells [160], retinoblastoma Y79 and WERI-Rb1 cells [161], and B78-H1 melanoma cells [10].
Telomestatin	Shortening the telomere length, inducing both telomeric and non-telomeric DNA damage, reduction of *c-Myb*, impairing the maintenance of glioma stem cells state by inducing apoptosis [9,162].	Anti-tumor activity on BCR-ABL-positive leukemic cell lines OM9;22 and K562 [163] and neuroblastoma [164], enhanced chemosensitivity toward daunorubicin and cytosine–arabinoside in acute myeloid leukemia cells [165].
Pyridostatin	Stabilizing the G-quadruplex [166], inhibiting telomerase activity and uncapping human POT1 from the telomeric G-overhang [167], eliciting a DNA damage response by causing the formation of DNA double strand breaks [128,168].	Enhanced chemosensitivity toward Olaparib-resistant *Brca1*-deleted tumor cells [168].
CX3543	Stabilizing the G-quadruplex, and disrupting nucleolin/rDNA G-quadruplex complexes in the nucleolus [169].	Anti-tumor activity in murine xenograft models of multiple human cancers, including breast (MDA-MB-231), pancreatic (MIA PaCa-2) [169].
CX5461	Inhibiting the initiation stage of rRNA synthesis and inducing both senescence and autophagy [170], blocking replication forks and inducing ssDNA gaps or breaks [147].	Anti-tumor activity in BRCA deficient cancer cells and polyclonal patient-derived xenograft models, including tumors resistant to PARP inhibition [147]. CX-5461 is now in advanced phase I clinical trial for patients with BRCA1/2 deficient tumors (Canadian trial, NCT02719977, opened May 2016).

**Table 2 molecules-24-00396-t002:** Proteins interacting with G-quadruplexes.

Types	Specific Proteins
Promoting/stabilizing proteins	Nucleolin, Topo1, thrombin
Unwinding proteins	Pif1, RHAU/DHX36, BLM, FANCJ, WRN, hnRNP A1/UP1, hnRNP D/BD2, XPD/XPB
Degrading proteins	GQN1, Mre11

**Table 3 molecules-24-00396-t003:** The different families of helicases.

Superfamily	Direction	Helicase
SF1	5′ to 3′, or 3′ to 5′	Pif1 [230,231,232], Dna2 [233,234]
SF2	5′ to 3′	Fe-S: FANCJ [26,27,235,236], DDX11 [237], RTEL1 [71,82]
	3′ to 5′	RecQ: BLM [238,239,240,241,242], WRN [243,244,245], Yeast Sgs1 [246,247]
SF3	3′ to 5′	SV40 T-antigen [248]
SF4	5′ to 3′	Twinkle [249]
SF5	5′ to 3′	RHAU [19,31,250,251,252,253]
SF6	5′ to 3′, or 3′ to 5′	mini chromosome maintenance (MCM) complex [254]

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
