# Peer review of "Developing Novel G-Quadruplex Ligands: From Interaction with Nucleic Acids to Interfering with Nucleic Acid–Protein Interaction"

_molecules, 2019, doi:10.3390/molecules24030396_

Round 1

Reviewer 1 Report

Zhi-Yin Sun et al. present a review paper named “Developing Novel G-Quadruplex Ligands: from Interaction with Nucleic Acids to Interfering with Nucleic Acid-Protein Interactions”.

As it is a review paper, one cannot expect significant novelty in the presented data, but such review should present exhaustive and up-to-date overview of the topic. This is where it substantially fails. The most expected part involving the interplay of quadruplex ligands and protein-quadruplex interactions, highlighted also in the manuscript title, is represented by few sentences, usually one per paragraph discussing particular ligand, i.e. three in total, with one reference in each. As a result, even the manuscript title does not reflect very well the content of the manuscript and I would recommend changing it.

The ligand part itself (lines 200-295) is highly “inspired” by David Monchaud’s review (ref. 78), including the sub-chapters and figures. This part contains only one reference (ref 77 – also review, by S. Neidle) dated after this Monchaud’s review, out of more than 40 used. In my opinion, the authors “inspired” themselves too much.

Concerning the quadruplex part, conformations schematically drawn in figure 2 are sometimes incorrectly assigned into groups. Some of the drawn conformations are not known (yet). At the end, it seems like the authors are not fully familiar with the quadruplexes.

I am not sure what the “folded topology” mentioned at lines 72 or 85, means in the context.

There are also several technical issues. The English should be significantly improved. There is quite a lot of typos and grammar mistakes in the manuscript, as well as inaccurate or misleading statements. Similarly, the chapter numbering is probably wrong and the organisation of chapters is misleading.

Next problem are the references: The location and usage of some references is questionable – ref 36 at 114; ref 42 at 137; etc. The authors very often interchange the first and last names of the reference authors. This is very confusing.

I do not think it is necessary to draw all three porphyrin ligands differing only in central metal ion in figure 6.

In general, I see no significant benefit in publishing the presented review, when compared to previous ones. Taking into account all the flaws mentioned above, I cannot recommend publishing the manuscript in Molecules in the present form without significant rewriting, in terms of both content and language.

I would recommend focusing more on “the ligands interfering with nucleic acid-protein interactions”, as it was probably the original aim of the manuscript, to cover in more details full range of such cases that were reported, and I would reduce the “not-very-well-done” quadruplex and ligand parts, much better reviewed elsewhere, to an indispensable minimum necessary for the main aim.

Author Response

1. The review should present exhaustive and up-to-date overview of the topic.

RESPONSEThanks for your advice. In this revision we pay particular attention to the introduction of recent advances in literature. A total of 39 literatures published in 2018 were added.

2. The most expected part involving the interplay of quadruplex ligands and protein-quadruplex interactions, highlighted also in the manuscript title, is represented by few sentences, usually one per paragraph discussing particular ligand, i.e. three in total, with one reference in each.

RESPONSEThanks for your suggestion. We realized that the original way of writing was very inappropriate. We reorganized this part into three parts to supplement the information, including G-quadruplexes ligands on quadruplex-related proteinsligands designing for blocking the protein-G-quadruplex interactions, and direct inhibitors for G-quadruplex-related proteins, and tried our best to make supplements on the basis of full literature review.

3. The ligand part itself (lines 200-295) is highly inspiredby David Monchauds review (ref. 78), including the sub-chapters and figures. 

RESPONSEThanks for your opinion. We realize that the original way of writing was very undesirable and unacceptable, thus, we re-wrote this part. In the revised manuscript, we focus on the five representative compounds with outstanding biological activity and the optimized compounds further developed basing on them, instead of the structure-based classification of the compounds. All the figures about compounds’ structures were redrawn (Figure 4 and 5).

4.  Concerning the quadruplex part, conformations schematically drawn in figure 2 are sometimes incorrectly assigned into groups. Some of the drawn conformations are not known (yet).

RESPONSEThanks for your opinion. We redrew Figure 2 basing on the newly structures reported in the PDB database.

5. I am not sure what the folded topologymentioned at lines 72 or 85, means in the context.

RESPONSEThanks for your opinion. We've deleted this statement.

6. There is quite a lot of typos and grammar mistakes in the manuscript, as well as inaccurate or misleading statements. Similarly, the chapter numbering is probably wrong and the organisation of chapters is misleading.

RESPONSEThanks for your opinion and sorry for our carelessness! We’ve checked the manuscript and corrected all the typos and grammar mistakes.

7. The location and usage of some references is questionable. 

RESPONSEThanks for your opinion and sorry for our carelessness! We’ve checked all the references and modified this error.

8. I do not think it is necessary to draw all three porphyrin ligands differing only in central metal ion in figure 6.

RESPONSEThanks for your advice. All the figures about compounds’ structures were redrawn and condensed into two diagrams. (Figure 4 and 5).

9. I would recommend focusing more on the ligands interfering with nucleic acid-protein interactions, and I would reduce the not-very-well-donequadruplex and ligand parts.

RESPONSEThanks for your advice. We simplified this part and tried to focus on the five representative compounds with outstanding biological activity and the optimized compounds further developed basing on them, instead of the structure-based classification of the compounds. 

Reviewer 2 Report

In this review, the authors describe the review of developing novel G-quadruplex ligands, from interacting with nucleic acids to interfering with nucleic acid-protein interaction.

G-quadruplex has a characteristic structure and biological function, such as in replication and transcription.The interactions between G-quadruplex and its ligand are summarized into many categories, and discussions are performed based on many references. Therefore, I recommends accepting after a minor revision.

Many small molecules that interact with G-quadruplex have been reported. Also, the review describes the formation of both parallel and anti-parallel G-quadruplexes. Authors should summarize these lists such as ligand, target G-quadruplex formation, and dissociation constant in a tabular form.

In this article, there are many descriptions concerning the inhibition of interactions between helicase and nucleic acid, but to attribute it as a protein in the title, the interaction of G-quadruplex with other proteins should also be described.

The position of the caption in Figure 1 should be corrected.

Author Response

1.     Authors should summarize these lists such as ligand, target G-quadruplex formation, and dissociation constant in a tabular form.

RESPONSEThanks for your advice. We’ve added a table to list typical G-quadruplex ligands and their biological activities and in vivo activities (Table 1). 

2.     The interaction of G-quadruplex with other proteins should also be described.

RESPONSEThanks for your opinion. We’ve added the interaction of G-quadruplex with other proteins, including the promoting/stabilizing proteins (such as nucleolin, topoisomerase, and CNBP), and degrading proteins (such as GQN1 and Mre11).

3.    The position of the caption in Figure 1 should be corrected.

RESPONSEThanks for your advice. We've changed the position of the caption in Figure 1 (Line 75). 

Reviewer 3 Report

Ou and co-workers focused on the control of interaction of G-quadruplexes and protein under the presence of G-quadruplex ligands. 

This reviewer feels that the present review collects relatively old papers. Currently, more excellent reviews come out with the similar subjects (for examples, Cell Press, “G-quadruplex: A Regulator of Gene Expression and Its Chemical Targeting” by Xiang Zhou et al., Nature review, “DNA G‐quadruplexes in the human genome: detection, functions and therapeutic potential” in 2017 by Shankar Balasubramanian and et al.). In addition, some important papers related the protein interfering by G4 are lacking (Paeschke, K. et al. Pif1 family helicases suppress genome instability at G‐quadruplex motifs. Nature 497, 458–462 (2013). , Mendoza, O., Bourdoncle, A., Boulé, J. B., Brosh, R. M. Jr & Mergny, J.‐L. G‐Quadruplexes and helicases. Nucleic Acid. Res. 44, 1989–2006 (2016). 

The G4 ligands discussed in this review are also relatively old.  Authors are strongly recommended to update more recent progress.

Author Response

This reviewer feels that the present review collects relatively old papers. In addition, some important papers related the protein interfering by G4 are lacking.Authors are strongly recommended to update more recent progress.

RESPONSEThanks for your advice. In this revision we pay particular attention to the introduction of recent advances in literature. A total of 39 literatures published in 2018 were added.

Round 2

Reviewer 1 Report

The authors present revised version of the manuscript named “Developing Novel G-Quadruplex Ligands: from Interaction with Nucleic Acids to Interfering with Nucleic Acid-Protein Interactions”.

Most importantly, the authors significantly expanded the most interesting part involving the interference of small molecules with the interaction of G-quadruplexes with proteins, mostly helicases. Moreover, much recent literature was referenced and the number of references significantly increased. In addition, the authors expanded also the parts with the G-quadruplex-binding proteins and especially with the G4 resolving helicases. The G4 ligand summary was also well reorganized and reduced to more clearly arranged form. Figures and tables were changed accordingly as well.

My only point remaining point aims towards the text and language quality. Although the authors claim “We’ve checked the manuscript and corrected all the typos and grammar mistakes.“, there is still quite a number of grammar errors and mistakes, including some misunderstandings in references (first vs last names – refs 134 and 136).

After correcting these minor points, I would recommend publishing the manuscript in Molecules.

Author Response

Dear reviewer, 

    Thank you for your suggestion! I am deeply sorry that we were not thoughtful enough in quoting literatures. We went over the references again to make sure that the important references had been cited.